


# Accounting for surface reflectance spectral features in TROPOMI methane retrievals

Alba Lorente[1], Tobias Borsdorff[1], Mari C. Martinez-Velarte[1], and Jochen Landgraf[1]

[1]Earth science group, SRON Netherlands Institute for Space Research, Leiden, the Netherlands

**Correspondence:** Alba Lorente (a.lorente.delgado@sron.nl)

**Abstract.** Satellite remote sensing of methane ($CH_4$) using the TROPOMI instrument is key to monitor and quantify emissions globally. In the past years, analysis of TROPOMI methane data has pointed to few false methane anomalies that can potentially be misinterpreted as enhancements due to strong emission sources. These artefacts are caused by spectral features of the underlying surfaces, which are not well represented in the forward model. Surface reflectance spectral dependence in the full-physics RemoTeC retrieval algorithm is modelled using a second order polynomial in wavelength. We show in this study that a third order polynomial better represents the surface reflectance dependency with wavelength of specific surface materials (e.g., rock), resulting in an improved characterization of the spectral features that caused the artificial localized $XCH_4$ enhancements found in several locations like e.g., Siberia, Australia, and Algeria. The use of a third order polynomial removed these artificial $XCH_4$ enhancements and significantly improved the fit over these specific features, while outside of these areas globally the fit did not improve in most cases. This reflects that a second order polynomial is optimal to capture the spectral dependencies of most surfaces given the characteristic of the TROPOMI instrument, but a third order polynomial is needed for the specific spectral characteristics of several surfaces. Furthermore, increasing the order of the polynomial to higher degrees did not further improve the retrieval. We also found that the known bias in retrieved methane for low albedo measurements slightly improves, but still a posterior correction needs to be applied, leaving open the question about the root cause of the albedo bias. After applying the third order polynomial globally, we perform the routine validation with TCCON and GOSAT. GOSAT comparison does not significantly improve, while TCCON validation results show an overall improvement of 2-4 ppb, reflecting that TCCON stations are not close to any of the corrected artefacts and are typically located around spectrally smooth surfaces.

## 1 Introduction

Methane ($CH_4$) obtained from TROPOMI measurements have been crucial to monitor and quantify methane emissions world-wide at global, regional and local scales (e.g., Qu et al., 2021; Chen et al., 2022; Sadavarte et al., 2021) and from multiple sources (e.g., Maasakkers et al., 2022; Shen et al., 2022; Lunt et al., 2019). The methane operational retrieval algorithm has provided data with high quality since the satellite instrument was launch in 2017 (Sha et al., 2021). The algorithm has been improved in the last years to better correct for biases from low albedo measurements, to update the spectroscopy and the regularization in the inversion, and to retrieve methane from measurements over the ocean under sun-glint geometry (Lorente et al., 2021, 2022a).





However, intercomparison between different scientific retrieval algorithms (Methane+ project, Lorente et al., 2022b) and the systematic analysis of TROPOMI methane data has pointed to biases linked to surface features that lead to false methane anomalies or artefacts (e.g., Barré et al., 2020). These artefacts can be misinterpreted as emissions caused by strong methane super emitters if they are found in e.g., active oil and gas areas, or due to processes related to global warming (e.g., Froitzheim

et al., 2021). In order to understand and to have an accurate quantification of methane emissions, it is important to correct for these artefacts on algorithm level and not just as a posterior correction. Recently, Jongaramrungruang et al. (2021) analyzed the impact of spectrally complex features on retrieved methane abundances, and how they could be mitigated based on the choices related to instrument and retrieval parameters. Based on the fact that different type of surface materials have specific spectral dependencies of their surface albedo, they showed that using larger order polynomials could reduce the retrieval biases related

to surface features.

In this study we show that increasing the order of the polynomial that models the surface reflectance spectral dependence in the TROPOMI methane retrieval algorithm removes large localized methane artefacts and improves the resulting spectral fit. We analyse the impact on global methane abundances and asses the quality of the dataset after applying the improved scheme to 4 years of TROPOMI measurements. After applying the third order polynomial globally, we perform the routine validation

with TCCON and GOSAT

## 2  Surface reflectance in RemoTeC

The methane total column-averaged dry-air mole fraction (XCH$_4$) is retrieved from TROPOMI measurements ($\boldsymbol{y}$) of sunlight backscattered by the Earth's surface and the atmosphere in the near-infrared (NIR, 757-774 nm) and shortwave-infrared (SWIR, 2305-2385 nm) spectral bands. We use the RemoTeC full-physics algorithm (described in detail by Hu et al. (2016) and Lorente

et al. (2021)) that simultaneously retrieves the amount of atmospheric methane and the physical scattering properties of the atmosphere. The forward model ($\boldsymbol{F}$) employs the LINTRAN V2.0 radiative transfer model in its scalar approximation to simulate atmospheric light scattering and absorption in a plane parallel atmosphere (Schepers et al., 2014; Landgraf et al., 2001)

The retrieval algorithm aims to find the state vector $\boldsymbol{x}$ that contains CH$_4$ partial sub-column number densities by solving the

minimization problem:

$$\hat{\boldsymbol{x}} = \min_{\mathrm{x}}\left(||\boldsymbol{S}_y^{-1/2}(\boldsymbol{F}(x)-\boldsymbol{y})||^2 + \gamma||\boldsymbol{W}(\boldsymbol{x}-\boldsymbol{x}_a)||^2\right), \tag{1}$$

where $||\cdot||$ describes the Euclidian norm, $\boldsymbol{S}_y$ is the measurement error covariance matrix that contains the noise estimate, $\gamma$ is the regularization parameter, $\boldsymbol{W}$ is a diagonal unity weighting matrix and $\boldsymbol{x}_a$ is the a priori state vector (Hu et al., 2016). The retrieval state vector contains CH$_4$ partial sub-column number densities at 12 equidistant pressure layers, the total columns of

the interfering absorbers CO and H$_2$O, the effective aerosol total column, size and height parameter of the aerosol power law distribution, and the spectral shift and fluorescence in the NIR band. A Lambertian surface albedo in both the NIR and SWIR spectral ranges and their spectral dependence as a polynomial is also retrieved.





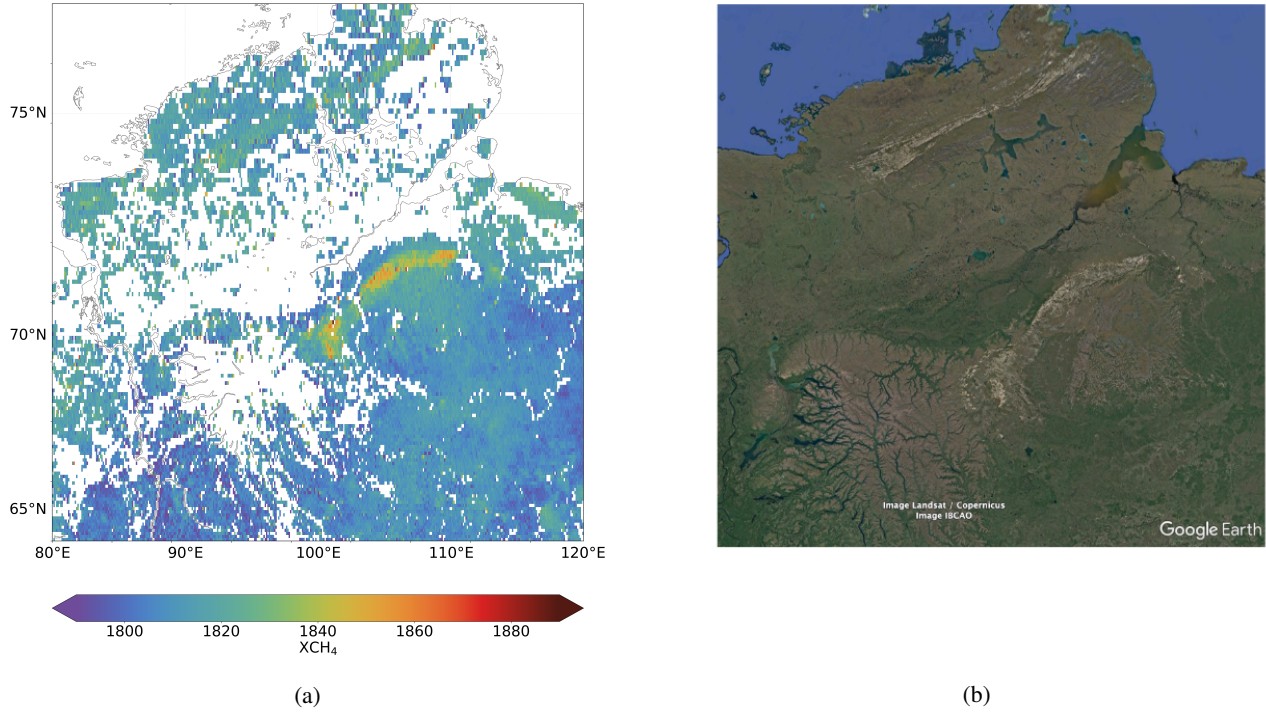

(a)  (b)

**Figure 1.** (a) TROPOMI XCH$_4$ retrieved over north Siberian lowland region (64-77 N, 80-120 E) averaged from March 2019 - March 2020 to a 0.1° x 0.1° grid and (b) Landsat imagery provided by © Google Earth ©Google, 2022.

Surface reflectance spectral dependence in RemoTeC is modelled using a low-order polynomial in wavelength. The coefficients of the polynomial are elements of the atmospheric state vector to be inferred in the inversion (Eq. 1). In the first version of the retrieval algorithm a second order polynomial was selected (Hu et al., 2016), but for specific surfaces this representation is not optimal and retrieved XCH$_4$ show enhancements strongly correlated with surface features. Figure 1 shows an example of an artificial XCH$_4$ enhancement over Siberia where high XCH$_4$ values coincide with a feature of outcrops of Paleozoic carbonate rocks (Froitzheim et al., 2021), recognizable in the Landsat imagery provided by Google Earth. Here, the surface is a specific type of carbonate rock for which the reflectance has a characteristic spectral feature. Even though this artefact can be mistaken by enhancements produced by underlying methane emissions (Froitzheim et al., 2021), the fact that these enhancements are constant in time and with a shape that does not change with meteorological (e.g., wind) conditions, supports the hypothesis that it is an artefact, as reported by Barré et al. (2020).

To improve the characterization of the surface reflectance dependency with wavelength in RemoTeC, following Jongaram-rungruang et al. (2021), we look at the ECOSTRESS spectral library (Meerdink et al., 2019), which contains laboratory spectra of minerals, rocks and man-made materials. Analysing the spectral information for different type of surfaces, particularly for rock and concrete, we find that a quadratic function might not be the most optimal representation of the surface reflectance spectral dependencies in the SWIR range. We thus increase the order of the polynomial and analyse the modelled radiance and



residuals for the particular case of the artefact over Siberia and in other areas where we see localized enhancements. We find that using a third order polynomial is enough to capture the specific spectral features that cause the localized enhancements (Sect. 2.1.1). The third order polynomial significantly improves the fit over the artificial enhancements. However, outside of these areas globally the fit does not improve in most cases, reflecting that a second order polynomial is optimal to capture the spectral dependencies of most surfaces (Sect. 2.1.2). Increasing the order of the polynomial to degrees higher than 3 does not further improve the retrieval in any of the cases, producing artificial signals in some of the retrieved parameters and worsening the quality of the fit outside of the specific areas.

## 2.1 Results

We retrieve four years of $XCH_4$ from TROPOMI measurements using a cubic function (i.e., third order polynomial) to better characterize the surface reflectance spectral features. This dataset corresponds to version 19_446 of the SRON scientific algorithm (see Data availability). In the operational processing, this feature was implemented in the processor version 02.04.00 in July 2022. In this section we show that with the updated treatment of surface reflectance several localized features like the one in Siberia (Fig. 1) are removed, and we also analyse the global effect and the consequences for the surface albedo bias described in Lorente et al. (2021).

### 2.1.1 Localized features

Figure 2a shows the SWIR radiance modelled with the forward model using a quadratic and a cubic function to characterize the spectral features of the surface reflectance for one of the pixels that show strong $XCH_4$ enhancement over Siberia (Fig. 1). The residuals (i.e., difference between modelled and measured radiance) for this specific pixel using a quadratic function show a strong dependency with wavelength (Fig. 2b), while for a pixel outside of the enhancement the dependency is not significant (not shown). If we increase the order of the polynomial from second to third order (i.e., using a cubic function), the dependency with wavelength of the residuals is significantly reduced (Fig. 2c). For this specific pixel, retrieved $XCH_4$ is reduced from 1885 ppb to 1847 ppb, and the $\chi^2$ value for the fit is reduced from 53 to 19 when going from second to third order polynomial.

Figure 3 shows three examples of regional TROPOMI $XCH_4$ for which using the third order polynomial removes artificial $XCH_4$ enhancements. The first row shows the effect over Siberia. Fig. 3a shows that TROPOMI $XCH_4$ retrieved using the third order polynomial does not reproduce the strong enhancement shown in Fig. 1. The differences between $XCH_4$ retrieved with the second and third order polynomial (Fig. 3b) show a strong decrease of retrieved $XCH_4$ over the outcrop of carbonate rocks. The average difference in $XCH_4$ is greater than 1% over the artefact. The fit significantly improves over the specific feature (Fig. 3c), implying that a third order polynomial can better capture the surface reflectance spectral dependence of the underlying surface. The fact that the fit does not show an improvement outside of the distinguishing feature means that a second order polynomial is a good representation elsewhere.

The second row in Fig. 3 shows the example of an artefact in north-western Australia that is removed when using the cubic function (Fig. 3d, e). Over this particular region the underlying soil is composed of sedimentary carbonate (Petheram et al., 2018), and the $XCH_4$ enhancement is also strongly correlated with the retrieved aerosol optical depth (not shown) which points

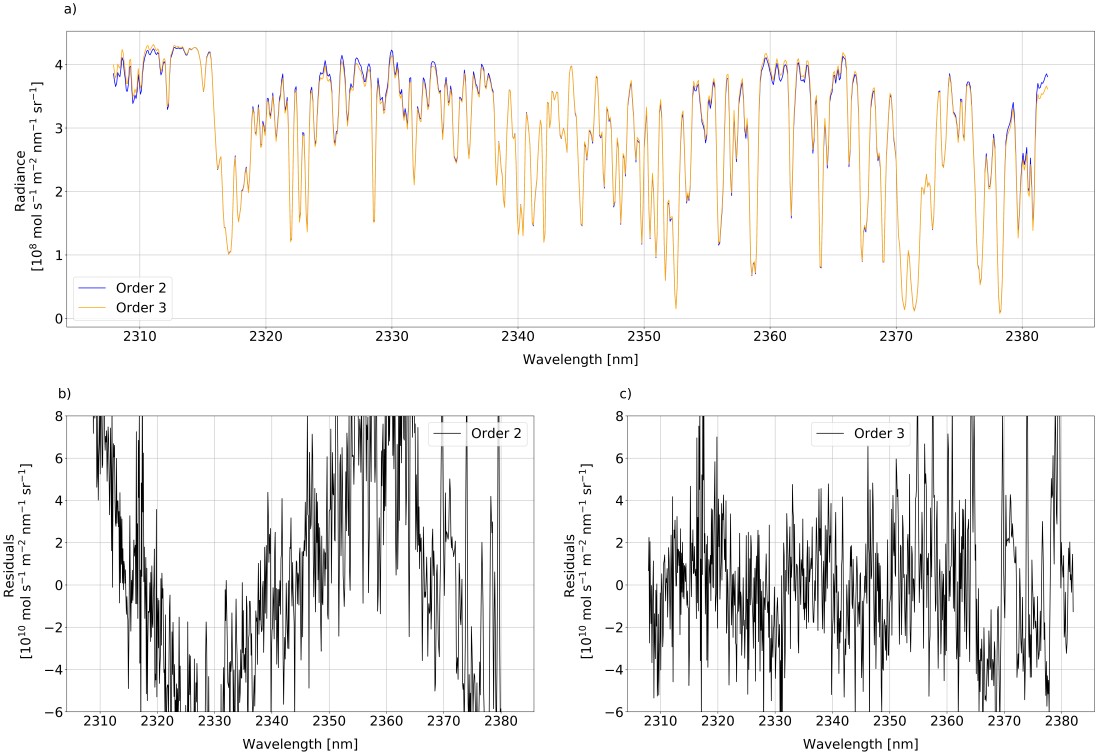

**Figure 2.** (a) Radiance modelled with RemoTeC with a second and third order polynomial to characterize the surface reflectance spectral features and (b, c) the respective residuals (modelled minus measured radiance) for a pixel (523369) in orbit 9147 over Siberia.

to an interference between the surface spectral features and the representation of scattering processes in the forward model. This XCH$_4$ enhancement could be misinterpreted as caused by emissions due to coal mining in the area. However, applying a third order polynomial removes the localized XCH$_4$ enhancement, and the fit significantly improves over the distinct feature (Fig. 3f).

The third row in Fig. 3 shows the example of multiple artefacts over north-western Algeria and eastern Tunisia that are removed when using the third order polynomial. Over this region, land cover is mostly bare rock and soil with some sparse grassland in the north. XCH$_4$ artefacts are correlated with different type of surfaces, some of them being rock and soil composed mostly of carbonate and limestone minerals (Jones et al., 2013). Methane emissions from the oil and gas sector have been actively monitored over this regions using TROPOMI data, so removing the artefacts is extremely relevant to correctly pinpoint

super emitters. The strongest effect shown in the difference $\Delta$XCH$_4$ plot in Fig. 3h does not occur over the major gas fields in



the area, so emissions previously quantified with TROPOMI over these fields are not affected. As in the previous two examples, the fit significantly improves over the specific features (Fig. 3i).

### 2.1.2 Global effect

In this section we analyse at regional and global scales the effect of using a third order polynomial to characterize surface
spectral features. Figure 4 shows over Asia XCH$_4$ retrieved with a second and third order polynomial, their ratio and the SWIR surface albedo. In the ratio plot (Fig. 4c) localized artefacts that are removed are clearly visible, for example in several points in the Tibetan Plateau, and also over Siberia as discussed in the previous section. Overall, the retrieved XCH$_4$ is more homogeneous, particularly over high latitudes. Over the low surface albedo band around the 60 N latitude region, retrieved XCH$_4$ is higher using a third order polynomial function.

Similar effects can be seen over North America (Fig. 5). Localized artefacts on the northern part of the Delaware Basin are removed when using a third order polynomial: this is relevant in order to attribute sources to individual locations in such large basins as the Permian using TROPOMI observations. Over the Interior Plains of Canada and the area surrounding the Grand Lakes, which are characterized by low surface albedo in the SWIR spectral band, retrieved XCH$_4$ is higher using a third order polynomial function, as well as in the western part of the United States.

The increase in retrieved XCH$_4$ over these low albedo regions when using a third order polynomial to characterize surface spectral feature reduces the low XCH$_4$ bias for which we apply a posterior correction. However, there is still a dependence with albedo, so we apply the "small-area approximation" as in Lorente et al. (2021) but using XCH$_4$ retrieved with the updated configuration. This implies that even the correction for scenes with low albedo is weaker, there is still an error source which causes the albedo biases that needs to be further investigated.

## 2.2 Validation


To assess the overall quality of the dataset, we perform the routine validation following (Lorente et al., 2021, 2022a) using ground based measurements from the TCCON network and measurements from the GOSAT satellite.

### 2.2.1 TCCON

We validate the TROPOMI XCH$_4$ dataset with ground-based measurements from the Total Carbon Column Observing Network
(TCCON) (Wunch et al., 2011) (data version GGG2014, downloaded on 15 Dec 2021). We collocate TROPOMI XCH$_4$ with a spatial radius of 300 km around each station, and a temporal overlap of 2 hours for the ground-based measurement and the satellite overpass. We average TROPOMI XCH$_4$ and compare it to the TCCON XCH$_4$, and for all individual paired collocations we estimate the mean bias of TROPOMI-TCCON XCH$_4$ differences and its standard deviation. We then compute the average of the biases of all stations and its standard deviation as a measure of the station-to-station variability as a diagnostic parameter
for the regional bias, following the approach in Lorente et al. (2021).



**Figure 3.** (a, d, g) TROPOMI XCH$_4$ retrieved with the third order polynomial to characterize the surface spectral features, (b, e, h) difference between XCH$_4$ retrieved with second and third order polynomial and (c, f, i) difference between $\chi^2$ value for the fit with second and third order polynomial over (first row) Siberia lowland region (64-77 N, 80-120 E), (second row) north-west Australia (11-23 S, 129-148 E), and (third row) north-west Algeria and east Tunisia (29-35 N, 3-10 E), averaged from March 2019 - March 2020 to a 0.1° x 0.1° grid.



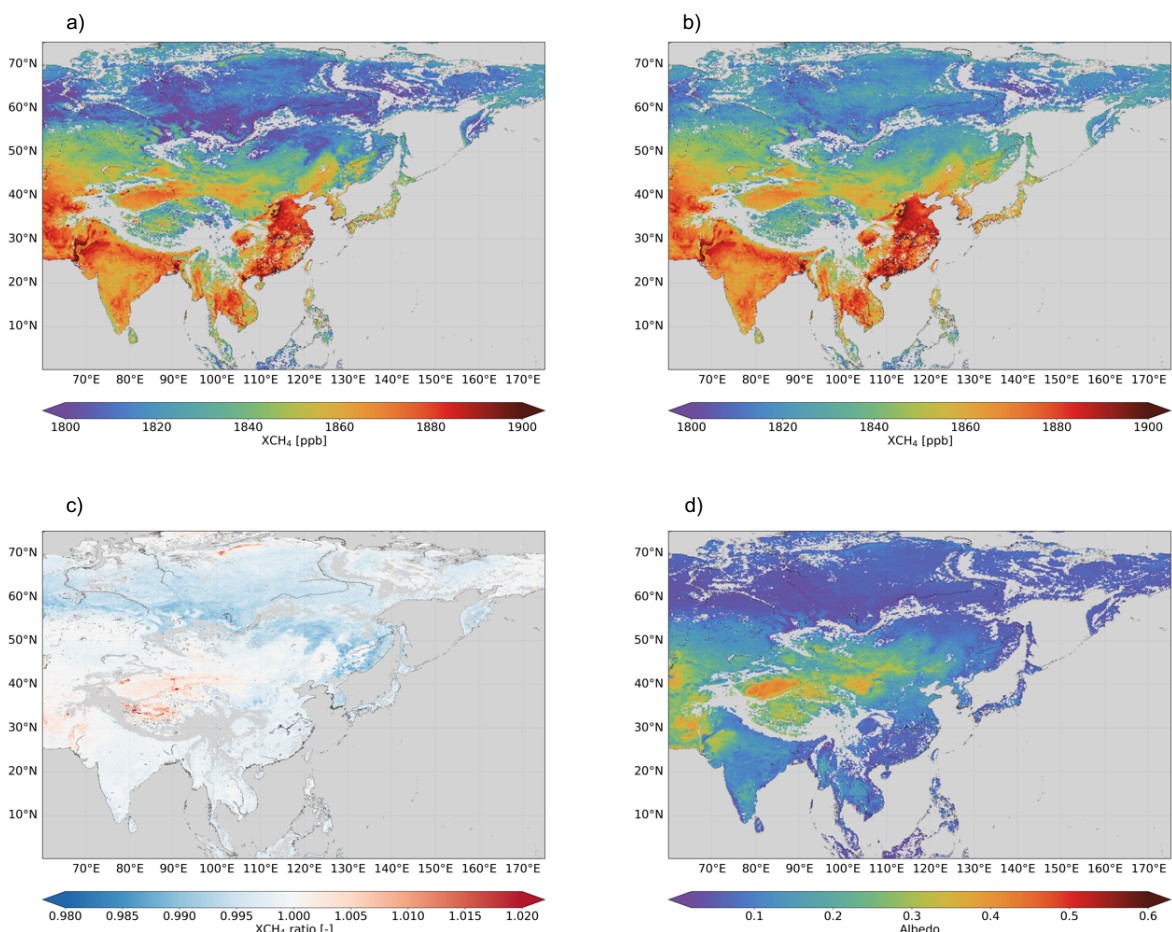

**Figure 4.** TROPOMI XCH$_4$ retrieved with (a) second and (b) third order polynomial to characterize the surface spectral features, (c) the ratio between XCH$_4$ retrieved with second and third order polynomial and (d) the retrieved surface albedo in the SWIR. Daily means in a 0.2° x 0.2° grid are averaged from Sept 2018 - Sept 2020



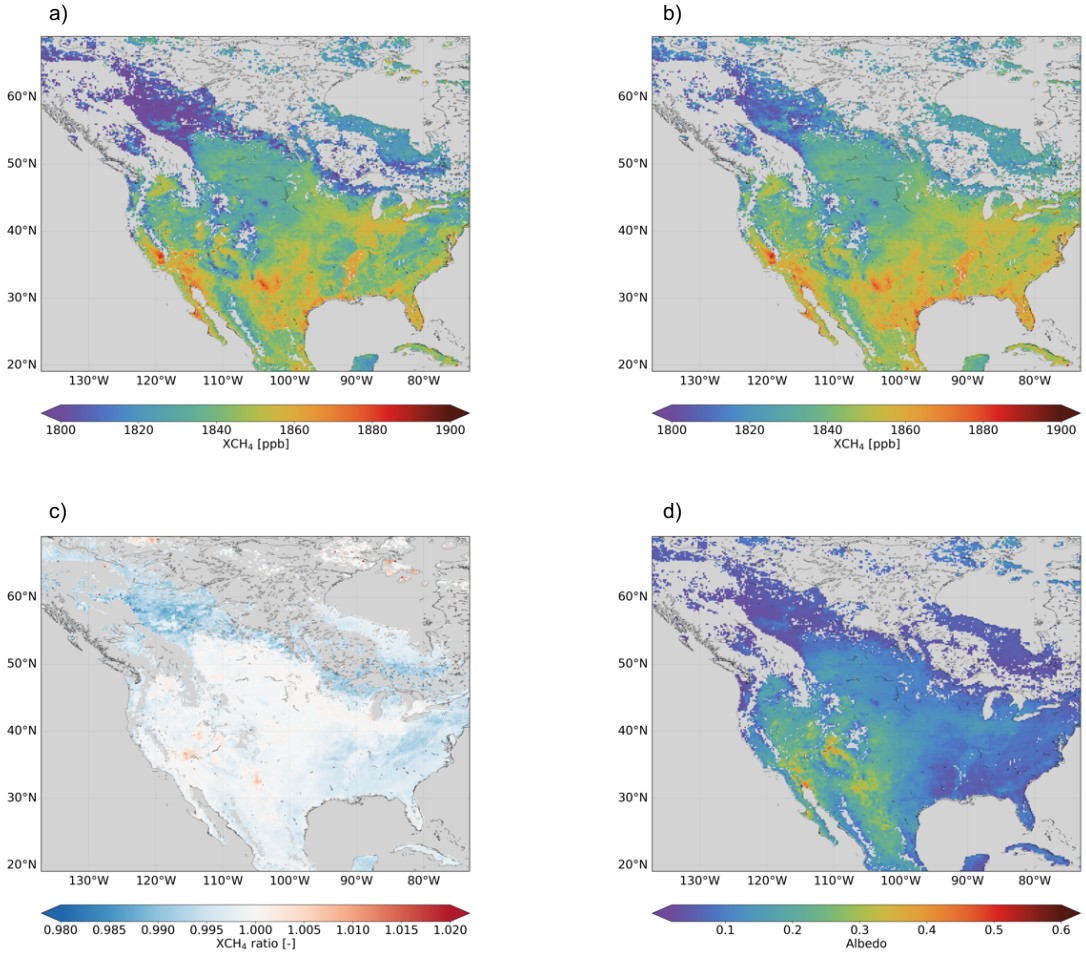

**Figure 5.** TROPOMI $XCH_4$ retrieved with (a) second and (b) third order polynomial to characterize the surface spectral features, (c) the ratio between $XCH_4$ retrieved with second and third order polynomial and (d) the retrieved surface albedo in the SWIR. Daily means in a $0.2°$ x $0.2°$ grid are averaged from Sept 2018 - Sept 2020





The mean bias is below 1 % for all stations; the validation results are summarized in Table 1. The average bias for all stations is -0.2 % ($-5.3$ ppb) and the station to station variability is 0.3% (5.1 ppb), both parameters below the mission requirements for TROPOMI $XCH_4$ retrievals. Compared to the uncorrected TROPOMI $XCH_4$, the mean bias is reduced from $-0.8$ % to 0.3 %. Figure 6a shows the mean bias and the standard deviation for each of the stations and Fig. 6b shows the correlation plot.

These validation results are of similar magnitude to the validation of TROPOMI $XCH_4$ retrieved using a second order polynomial (Lorente et al., 2022a). TCCON suggests an improvement of around 2-4 ppb in the bias and station-to-station variability when increasing to a third order polynomial. As the biggest changes with the new configuration occur at very localized scales, the improvement in the validation is small compared to the $XCH_4$ decrease over the enhancements. TCCON stations are typically located over homogeneous and spectrally smooth surfaces, and none of the stations is close to any of the

detected artefacts and therefore cannot capture the most significant changes in retrieved $XCH_4$.

**Table 1.** Overview of the validation results of TROPOMI $XCH_4$ land measurements with measurements from the TCCON network at selected stations. The table shows number of collocations, mean bias and standard deviation for each station and the mean bias for all stations and the station-to-station variability. Results are shown for TROPOMI $XCH_4$ with and without the albedo bias correction applied.

| Site, Country, Lat-Lon Coord. | Nr. of points | Corrected TROPOMI $XCH_4$ and TCCON | | Uncorrected TROPOMI $XCH_4$ and TCCON | |
|---|---|---|---|---|---|
| | | Bias [ppb] (%) | Standard deviation [ppb] (%) | Bias [ppb] (%) | Standard deviation [ppb] (%) |
| **Pasadena** (US) (34.14, $-118.13$) | 699 | $-5.5$ ($-0.3$) | 9.3 (0.5) | $-1.0$ (0.0) | 9.3 (0.5) |
| **Saga** (Japan) (33.24, 130.29) | 276 | 6.6 (0.3) | 13.4 (0.7) | $-10.1$ ($-0.5$) | 13.7 (0.7) |
| **Karlsruhe** (Germany) (49.1, 8.44) | 295 | $-3.5$ ($-0.2$) | 9.5 (0.5) | $-15.8$ ($-0.9$) | 10.0 (0.5) |
| **Darwin** (Australia) ($-12.46$, 130.93) | 198 | $-10.7$ ($-0.6$) | 12.9 (0.7) | $-17.6$ ($-1.0$) | 13.4 (0.7) |
| **Wollongong** (Australia) ($-34.41$, 150.88) | 423 | $-6.4$ ($-0.4$) | 11.5 (0.6) | $-11.7$ ($-0.6$) | 11.6 (0.6) |
| **Lauder II** (New Zealand) ($-45.04$, 169.68) | 358 | $-2.9$ (0.1) | 11.3 (0.6) | $-13.2$ ($-0.7$) | 11.2 (0.6) |
| **Park Falls** (US) (45.94, -90.27) | 582 | $-7.2$ ($-0.4$) | 13.3 (0.7) | $-22.6$ ($-1.2$) | 16.1 (0.9) |
| **East Trout Lake** (Canada) (54.36, $-104.99$) | 491 | $-5.9$ ($-0.3$) | 14.8 (0.8) | $-21.8$ ($-1.1$) | 16.6 (0.9) |
| **Lamont** (US) (36.6, $-97.49$) | 664 | $-10.4$ ($-0.6$) | 8.7 (0.5) | $-15.8$ ($-0.8$) | 9.7 (0.5) |
| **Orléans** (France) (47.97, 2.11) | 390 | $-4.5$ (0.2) | 11.9 (0.6) | $-15.3$ ($-0.8$) | 13.8 (0.7) |
| **Edwards** (US) (34.95, $-117.88$) | 757 | 0.4 ($-0.0$) | 8.9 (0.5) | 4.1 (0.2) | 9.1 (0.5) |
| **Sodankylä** (Finland) (67.37, 26.63) | 356 | $-13.8$ ($-0.7$) | 17.3 (0.9) | $-34.2$ ($-1.8$) | 17.8 (1.0) |
| **Mean bias, station-to-station variability** | | $-5.3$ ($-0.3$) | 5.1 (0.3) | $-14.6$ ($-0.8$) | 9.5 (0.5) |





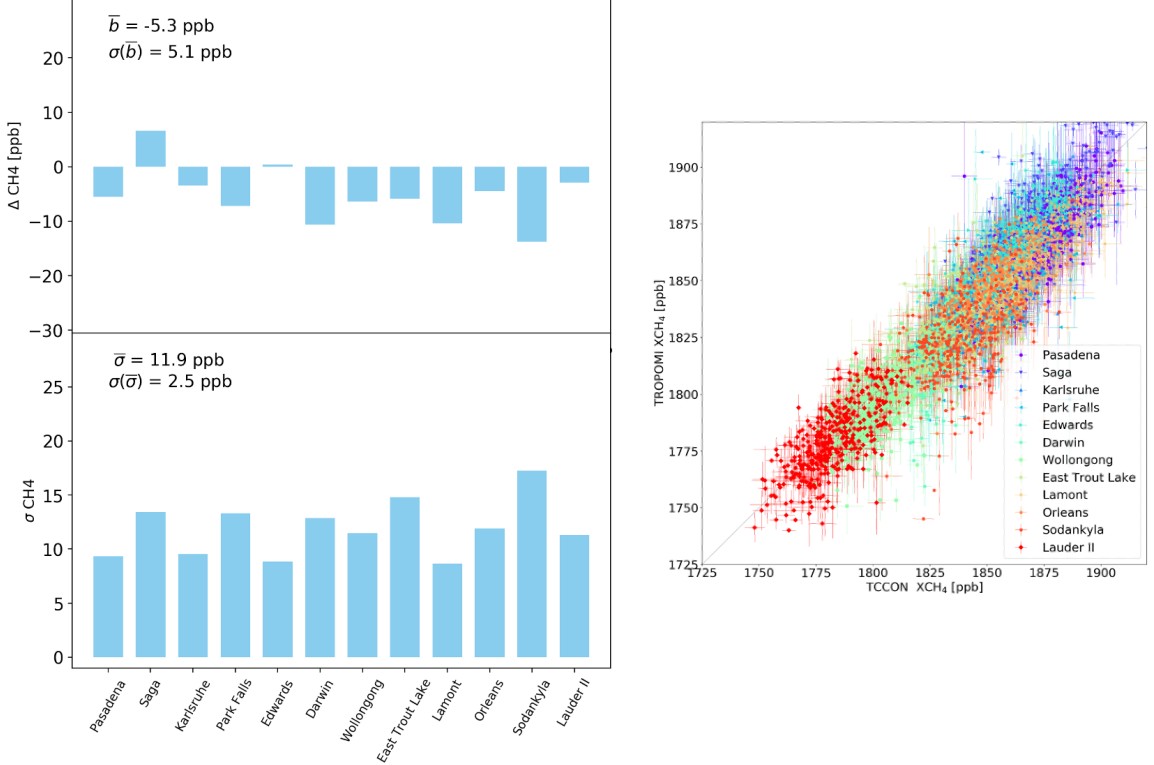

**Figure 6.** (a) Mean differences between TROPOMI and TCCON XCH$_4$ ($\Delta$XCH$_4$), and the standard deviation of the differences ($\sigma_{\mathrm{XCH}_4}$) for each of the stations selected for the validation. (b) Correlation of daily average XCH$_4$ measured by TROPOMI and TCCON for all the stations.

### 2.2.2 GOSAT

In this section we use XCH$_4$ measurements by The Thermal And Near infrared Sensor for carbon Observation - Fourier Transform Spectrometer (TANSO-FTS) on board the Greenhouse gases Observing SATellite (GOSAT) satellite for the validation of TROPOMI XCH$_4$ data. We use the GOSAT proxy XCH$_4$ data product produced at SRON in the context of the ESA GreenHouse Gas Climate Change Initiative (GHG CCI) project (Buchwitz et al., 2019, 2017). This XCH$_4$ product is retrieved using the RemoTeC/proxy retrieval algorithm.

We compare XCH$_4$ retrieved from TROPOMI and GOSAT measurements for the period of Mar 2018 - Dec 2020, and we compute the average of daily biases and its standard deviation between TROPOMI and GOSAT measurements gridded in a 2° x 2° grid. Globally on average TROPOMI XCH$_4$ underestimates GOSAT XCH$_4$. The comparison leads to a bias after correction of -14.4 $\pm$ 15.9 ppb ($-0.7 \pm 0.8$ %) and a Pearson's correlation coefficient of 0.87.

Similarly to the TCCON validation results, the comparison results for GOSAT are as well of similar magnitude to the comparison of TROPOMI XCH$_4$ retrieved using a second order polynomial (Lorente et al., 2022a). This again reflects the





fact that biggest changes in retrieved TROPOMI XCH$_4$ using a third order polynomial happen at localized scales. Due to the
GOSAT reduced coverage compared to that of TROPOMI and the coarse grid used in the comparison, the effect on retrieved
TROPOMI XCH$_4$ is not reflected in the TROPOMI to GOSAT comparison.

## 3   Conclusions

The aim of the study was to improve the characterization of the surface reflectance spectral features in the forward model
of the TROPOMI XCH$_4$ retrieval algorithm. In the last years, studies that analysed TROPOMI methane data as well as an
intercomparison between different scientific retrieval algorithms pointed to few false methane anomalies linked to features of
the underlying surfaces. These anomalies can be misinterpreted as caused by emission sources ((e.g., Froitzheim et al., 2021)).
We have shown here that increasing the order of the polynomial that models the surface reflectance spectral dependence in the
TROPOMI methane retrieval algorithm removes the large localized methane artefacts and improves the resulting spectral fit.

Surface reflectance spectral dependence in the full-physics RemoTeC retrieval algorithm is modelled using a low-order
polynomial in wavelength. Analysing the spectral information of distinct materials from the ECOSTRESS spectral library, we
found that a quadratic function (i.e., second order polynomial) was not optimal to represent the surface reflectance spectral
dependencies of surface materials such as rock and concrete in the SWIR spectral range. In the particular case of the artefact
over Siberia, where the underlying surface is composed mainly of carbonate rock, we find that the residuals for specific pixels
over the strong XCH$_4$ artefact have a strong dependency with wavelength. When increasing the order of the polynomial that
characterizes the surface reflectance spectral dependency in the forward model from second to third, this dependency of the
residuals was significantly reduced.

We have shown that using a third order polynomial improved the characterization of the spectral features that caused the
artificial localized XCH$_4$ enhancements found in several locations like e.g., Siberia, Australia, and Algeria. These artefacts were
located close to regions with known emissions due oil, gas and coal extraction activities, so they could be easily misinterpreted
as caused by "real" methane sources. The use of a third order polynomial removed these artificial XCH$_4$ enhancements and
significantly improved the fit over these specific features. Outside of the specific features, the spectral fit did not show such
substantial improvements, reflecting the fact that a second order polynomial is sufficient to capture the spectral dependencies of
most surfaces, given the characteristics of the TROPOMI instrument. We also tested increasing the order of the polynomial to
higher degrees, but the retrieved parameters showed undesired behaviour outside of the specific areas where the XCH$_4$ artefacts
were found, including a worsening of the spectral fit.

The analysis at regional and global scales of the effect of using the improved characterization of the surface spectral features
shows that over low surface albedo areas (e.g., region around the 60 N latitude in the asian continent), retrieved XCH$_4$ is higher
using a third order polynomial function. As a consequence, the known bias in retrieved methane for low albedo measurements
slightly improves, but still a posterior correction needs to be applied. This implies that even the correction for scenes with low
albedo becomes weaker, there is still an error source which causes systematic albedo biases that needs to be further investigated.



Finally we asses the quality of the dataset after applying the third order polynomial globally, performing the routine valida-
tion with TCCON and GOSAT. GOSAT comparison does not significantly improve, while TCCON validation results show an
overall improvement of 2-4 ppb, reflecting that TCCON stations are not close to any of the corrected artefacts and are typically
located around spectrally smooth surfaces.

*Data availability.*    The TROPOMI $CH_4$ dataset of this study is available for download at ftp://ftp.sron.nl/open-access-data-2/TROPOMI/tropomi/ch4/
version 19 446

*Author contributions.*    AL, TB, MCV, and JL provided the TROPOMI $CH_4$ retrieval and data analysis. AL wrote the original draft and all
authors discussed the results and reviewed and edited the paper.

*Competing interests.*    The authors declare that they have no conflict of interest.

*Disclaimer.*    The presented work has been performed in the frame of Sentinel-5 Precursor Validation Team (S5PVT) or Level 1/Level 2
Product Working Group activities. Results are based on preliminary (not fully calibrated or validated) Sentinel-5 Precursor data that will still
change. The results are based on S5P L1B version 1 data. Plots and data contain modified Copernicus Sentinel data, processed by SRON.

*Acknowledgements.*    The TROPOMI data processing was carried out on the Dutch National e-infrastructure with the support of the SURF
Cooperative. Funding through the TROPOMI national program from the NSO and Methane+ is acknowledged.



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
