# Peer review of "Accounting for surface reflectance spectral features in TROPOMI methane retrievals"

_Atmospheric Measurement Techniques, 2022_

## Author Comment (AC1)

**Response to the reviewers on the manuscript "Accounting for surface reflectance spectral features in TROPOMI methane retrievals" by Alba Lorente et al.**

The authors would like to thank the reviewers for their comments and suggestions. Below are the comments by the reviewers in blue and replies in black. Any modification made to the text is shown in italics. The line and page numbers correspond to the version of the manuscript available for online discussion.

**Reviewer 1**

**Comment C 1.1** — Lines 83-84 and in results in general: throughout the text, it is emphasized that the second-order polynomial is optimal for most surfaces, and the third-order polynomial is optimal for areas with surface artifacts even though it shows no improvement in the rest of the surfaces. After reading the text, it is not clear to me if, from now on, the operational product is the TROPOMI data processed fully with the third-order polynomial, if it is a combination of both polynomials depending on the area, or if the operational product is the data processed with the second-order polynomial and the third-order polynomial is provided separately but is not considered for the official product.

**Reply**: The retrieval algorithm uses a single polynomial, either a polynomial of second or third order, so there is no combination possible. From now on, the retrieval algorithm will use a third order polynomial everywhere, as it improves significantly the results.

In order to make the text clearer for the reader, we have simplified the text in several places throughout the manuscript to avoid repetition of '2nd order polynomial', '3rd order polynomial' and some other instances. Examples of these changes are in the abstract and in the results section lines 95-100 (110-115 in the tracked changes version). We have modified the text in lines 83-84 at the beginning of the Results section to highlight the implementation of the updates presented in the manuscript to the operational processing algorithm. We have added the following sentence in the conclusions, at the end of the third paragraph: "Based on these results, we apply the third order polynomial for the surface reflectance spectral dependency globally". We have also specified in data availability section that the use of the third order polynomial for the surface reflectance characterization was implemented in the operational processing chain in July 2022.

**Comment C 1.2** — Lines 115-116: The authors point out that, in the case of Algeria, the effects do not occur in the major field of the country. However, in Figure 3, the effect of surface artifacts can be seen in the Hassi R'Mel field, west of the image, or the Illizi basin, south of the image

(only the northern part of the basin is seen in the image, but the same type of surface extends over several kilometers). In these areas, the OG production is high, and, for example, Lauvaux et al. 2022 reported ultra-emissions detected with TROPOMI in these areas. Does this imply that the quantified emissions in these areas may have erroneous estimates?

If so, in the same way that the authors mention confidence in the data from the major field in Algeria, it would be convenient to also warn about the danger in the rest of the areas affected by the artifacts.

**Reply**: We acknowledge that this statement is vague and not supported by any evidence in the manuscript. Figure R1 shows the difference plot of XCH4 retrieved with the 2nd and 3rd order polynomial (same as Fig. 3, middle column) over Algeria for a greater spatial domain than Fig. 3 in the manuscript (27-35N vs. 29-35N, 2-10E vs. 3-10E). Mayor oil and gas fields from the country have been marked with 'x' on the map. The Hassi R'Mel field is just on the edge of one of the XCH4 artificial enhancements that has been corrected for, similarly as Tin Fouye, Ohanet and Alrar fields south of the image. In the case of emission plumes from these fields, if the wind is blowing towards the direction of the artifacts, indeed these emissions will, with high probability, be missed because the plume signal will be diffused into the false enhancement, and will be classified as an artefact by an automated detection algorithm like Berend et al. (2023). With the new retrieval scheme, plume-like structures will not be discarded because of high correlation with albedo due to the retrieval artefacts. Hassi Messaoud is another mayor oil and gas field where many plumes are detected, and this one is located in areas where the change in XCH4 is not significant, therefore emission estimates from that field in studies like Lauvaux et al. (2022) or Berend et al. (2023) are hardly affected.

In the case of regional emissions, the effect of the new retrieval scheme is difficult to quantify. Regional or country wide emissions estimates may be lower, as overall retrieved  $XCH_4$  is lower for Algeria. It could be that for areas where there are artifacts the inversion methods attributed more uncertainty when using the old retrieval scheme, therefore using the new retrieval scheme will at least provide more reliable emission quantification.

Based on this, we have removed lines 115-116 and modify the text according to the discussion above:

"The Hassi R'Mel field is just on the edge of one of the artefacts that has been removed with the new retrieval scheme.  $XCH_4$  signal from plumes that blow towards the false enhancement because of specific wind conditions might get diffused within the enhancement, and therefore most probably classified as artefacts by plume detection algorithms due to the high correlation with albedo. Hassi Messaoud is another mayor oil and gas field from which plumes are detected. It is located in areas where the change in  $XCH_4$  by the updated retrieval is not significant, therefore emissions estimates from the field should not be affected. In the case of area emission estimation, the effect of the updated retrieval will depend on how the artefacts and uncertainty on the observations are characterized on the inversion schemes. As overall retrieved  $XCH_4$  is lower for the country of Algeria, estimated regional emissions with the new retrieval scheme may also be lower."

Figure 1:  $XCH_4$  difference that shows  $XCH_4$  retrieved with second order polynomial minus  $XCH_4$  retrieved with third order polynomial, and the location of the mayor oil and gas fields in the country.

**Comment** C1.3 — Figure 3: I suggest adding a label on the left of each row indicating the country or region that is being represented. This is already indicated in the image caption, but I think adding the label in the figure could improve the visualization of the image.

**Reply**: We have added labels to each of the figures to make visualization clearer.

**Comment C1.4 — Technical corrections:**

Line 2: add "a" before "few false..." => pointed to a few false Done

- L14: add comma after "still" => but still, a posterior Done
- L19: measurements have => has Corrected
- L20: Add comma after "regional" => global, regional, and local Done
- L22: was launch => was launched Corrected
- L23: remove the second "to" => albedo measurements, update the... Done
- L24: remove "to" before "retrieve" => and retrieve methane Done
- L33: add s in "type" => different types Corrected
- L36: add comma after "study" => In this study, Done
- L38: asses => assess Corrected

L57: add comma after "matrix" = weighting matrix, and Done

L60: add comma after "algorithm" and "surfaces" => retrieval algorithm, a second order polynomial was selected (Hu et al., 2016), but for specific surfaces, this Done

L61: show => shows Corrected

L70: add comma after "rocks" => minerals, rocks, and Done

L84: add comma after "section" => In this section, we Done

L104: add comma after "region" => Over this particular region, the underlying Done

L105 ad comma after "shown)" => (not shown), which Done

L112: add s in "type" => different types Corrected

L114: this regions => these regions Corrected

L119: add comma after "section" => In this section, we Done

L120: add comma after "ratio" => their ratio, and Done

L121: add comma after "4c)" and "example" => plot (Fig. 4c), localized artefacts that are removed are clearly visible, for example, in several points Done

L122: add comma after "Siberia" => over Siberia, as discussed Done

L133: even the correction => even if the correction Corrected

Figure 4: add comma after "polynomial" in the second line => Done

Figure 5: add comma after "polynomial" in the second line => Done

L147: add comma after "(5.3 ppb)" = 0.2 % (-5.3 ppb), and Done

L149: add comma after "stations" => the stations, and Fig. 6b Done

L155: add comma before and after "therefore" => artefacts and, therefore, cannot Done

Table 1: add "the" before "number" and comma after "bias" and "stations" in the second line => The table shows the number of collocations, mean bias, and standard deviation for each station and the mean bias for all stations, and Done

L157: add comma after "section" => In this section, we Done

L164: add comma after "average" => on average, TROPOMI XCH4 Done

L168: add "the" before "biggest" => that the biggest Done

L 174-175: add comma after "data" and "algorithms" and add "a" before "few"  $=_{i}$  TROPOMI methane data, as well as an intercomparison between different scientific retrieval algorithms, pointed to a few false methane Done

L188: add "to" before "oil" and add comma after "gas" => emissions due to oil, gas, and coal Done

L196: Asian in capital letter Corrected

L198: add comma after "still" and "if" after "even" => but still, a posterior correction needs to be applied. This implies that even if the Done

L200: add comma after "Finally" and replace "asses" with assess Done

We thank the reviewer for the careful correction of the text, particularly the punctuation with the commas.

**Reviewer 2**

**Comment** C2.1 — Figures 1 and 3: Perhaps an inset in each row, showing the location of the zoom area in a global or at least regional context would be helpful here.

**Reply**: We acknowledge the advantage of having the zoom area in a global context. However, as these are very localized effects we prefer to have a big figure showing the artefact rather than having such an inset as this will take up much space of the figure. We have included a label referring to the exact location of each region, also as a response to Comment 1.3 by reviewer 1.

**Comment** C2.2 — Figure 3: make it clearer that the difference plots show the 2nd-order scheme minus the 3rd order scheme. I think that the phrase "difference between XCH4 retrieved with second and third order polynomial" is not quite clear enough.

**Reply**: We have changed the sentence to:  $(b, e, h) XCH_4$  difference that shows  $XCH_4$  retrieved with second order polynomial minus  $XCH_4$  retrieved with third order polynomial and (c, f, i) difference between  $\chi^2$  value for the fit with second minus third order polynomial [...]

**Comment** C 2.3 — Line 147 and elsewhere in that paragraph: I wasn't quite sure that the numbers in the text matched that in Table 1. For example, the text states that the average bias for the corrected XCH4 is -0.2% whereas the Table states that it is -0.3%. Then on line 149, the bias is stated as +0.3%. Otherwise, if I have misunderstood, please clarify the text.

**Reply**: We thank the reviewer for spotting these errors. The numbers in the text have been corrected according to what is shown in Table 1, which are the correct ones.

**Comment** C 2.4 — Line 148: I was initially confused about the use of the word 'uncorrected' here as I assumed that it was referring to use of the second-order polynomial. The caption text above Table 1 does make the meaning clear but it would helpful if this information was included in the main text here too.

**Reply**: We have added to line 148 the following text: (e.g., without the albedo bias correction applied). Before the validation section (lines 155-165 in the tracked changes version), we have also extended the discussion on the albedo bias correction acknowledging that some background information on this topic was missing.

**Comment C 2.5** — Line 167: Could you briefly state the second-order values here to aid the reader? "Similar magnitude" is a bit vague.

**Reply**: We have added the values together with the citation to the analysis Lorente et al. (2022). We have done the same in Line 166 for the GOSAT comparison results.

**Comment C 2.6** — General throughout text: You state that the third-order polynomial scheme significantly improves results over the regions that previously had artefacts due to errors in the surface reflectance, but no improvements away from these regions. You state that a second order-polynomial is optimal elsewhere, in fact. Does this mean that the third-order scheme leads to reduced performance elsewhere and the second-order scheme should be used there – i.e. a mix of schemes is necessary? Or just that the third-order scheme is used globally but does not produce improvements in most regions? The text discussing this needs to be clearer.

**Reply**: We acknowledge that this may cause some confusion. What we have tried to show in the results section is that the third order scheme doesn't lead to a reduced performance outside of the specific artefacts, therefore it can be applied globally. This is mainly shown by the spectral fit which significantly improves where the artefacts are removed but elsewhere is not reduced. Therefore, a mix of schemes is not necessary.

We have added the following sentence in the conclusions, at the end of the third paragraph: "Based on these results, we apply the third order polynomial for the surface reflectance spectral dependency globally."

We have significantly modified the abstract to state things in a more clear and direct way, removing and adding quite some text, and also throughout the results section to avoid repetition of '2nd order polynomial', '3rd order polynomial' and some other instances. Examples of these changes are lines 95-100 (110-115 in the tracked changes version).

**Comment C 2.7 — Technical corrections:**

- Line 19: have been -> has been Corrected
- Line 22: was launch -> was launched Corrected
- Line 33: type -> typesCorrected
- Line 38: asses  $\rightarrow$  assess Corrected
- Line 133: even the  $\rightarrow$  even though the Changed to 'even if the'
- Line 166: remove 'as well' Done